# Effect of Changes in Veterinary Feed Directive Regulations on Violative Antibiotic Residues in the Tissue of Food Animals from the Inspector-Generated Sampling in the United States

**DOI:** 10.3390/microorganisms10102031

**Published:** 2022-10-14

**Authors:** Shamim Sarkar, Chika C. Okafor

**Affiliations:** Department of Biomedical and Diagnostic Sciences, College of Veterinary Medicine, University of Tennessee, Knoxville, TN 37996, USA

**Keywords:** antibiotic, residues, violative, penicillin, tetracycline, florfenicol, VFD, sulfonamides

## Abstract

The presence of antibiotic residues in the tissue of food animals is a growing concern due to the adverse health effects that they can cause in humans, such as antibiotic resistance bacteria. An inspector-generated sampling (IGS) dataset from the United States National Residue Surveillance Program, collected between 2014 and 2019, was analyzed to investigate the association of changes in the veterinary feed directive (VFD) regulations on the detection of violative penicillin, tetracycline, sulfonamide, desfuroylceftiofur, tilmicosin, and florfenicol, residues in the tissue of food animals. Multivariable logistic regression models were used for analysis. While the animal production class was significantly associated with residue violations for tetracycline, having a sample collection date after the implementation of change in VFD regulations was not. However, the odds of detecting violative sulfonamide and penicillin residues in the tissue of food animals following the implementation of the change in VFD regulations were 36% and 24% lower than those collected before the implementation of the change in VFD regulations period, respectively, irrespective of animal production class. Violative desfuroylceftiofur, tilmicosin, and florfenicol residues in the tissue of food animals were not significantly associated with the implementation of changes in the VFD regulations. Further investigation of the factors that influence the presence of violative antibiotic residues in the tissue of food animals following the change in VFD regulations would lend clarity to this critical issue.

## 1. Introduction

Antibiotics have been widely used for the treatment, control, and prevention of livestock diseases in the United States (U.S.) [1,2,3]. Inappropriate use of antibiotics in food animals is one factor associated with the presence of violative antibiotic residues (ARs) in food animal products [4]. A prior study found violative tetracycline, gentamicin, oxytetracycline, and penicillin residues in bob veal calves in the U.S. [5]. Likewise, penicillin was the most frequently identified antibiotic with violative residue levels in culled cows in the U.S. [5]. Foods of animal origin containing ARs have adverse health effects among consumers. For example, ingestion of antibiotic-containing meat products can induce resistance in the normal flora of the human gastrointestinal tract [1]. 

The Veterinary Feed Directive (VFD) regulations were updated by the U.S. Food and Drug Administration (FDA) on 1 October 2015 and fully implemented on 1 January 2017 in accordance with FDA’s Guidance for Industry #213 [6]. This VFD rule change guideline discusses FDA’s concerns regarding the development of antimicrobial resistance in human and animal bacterial pathogens when medically important antimicrobial drugs are used in food-producing animals in an injudicious manner. So the modified VFD rule aims to promote the judicious use of medically important antimicrobials in food-producing animals in the U.S. [7]. The VFD rule changes restrict the use of medically important antimicrobials administered in feed and water for therapeutic purposes only and require the supervision of a licensed veterinarian [7]. However, a recent qualitative study reported that the VFD could create more black-market access to in-feed antimicrobials [8]. Previous studies reported increased use of antimicrobials for therapeutic purposes in food-producing animals after a rule restricting antimicrobial use (AMU) for growth promotion in food animals was implemented in Denmark and Sweden [9,10]. On the other hand, implementing a rule restricting AMU in Taiwan in 2000 was associated with decreased resistance to vancomycin among enterococci in chickens [11].

In 1976, the U.S. established the U.S. National Residue Program (NRP), a national residue surveillance system to monitor chemical residues, including antibiotic residues, in meat, poultry, and egg products. This surveillance program was aimed at protecting the health and welfare of consumers. The NRP is an interagency program conducted by the U.S. Department of Agriculture’s (USDA) Food Safety and Inspection Service (FSIS) [12]. The NRP has three sampling schemes: surveillance sampling, inspector-generated sampling (IGS), and unique project sampling [12]. The inspector-generated sampling targets individual suspect animals, suspect animal populations, and animals retained or condemned for specific pathologies. The following steps are involved in the inspector-generated sampling: a Public Health Veterinarian (PHV) selects a carcass for sampling based on the criteria (i.e., an animal with disease signs and symptoms, producer history of violative levels of residues, or as a follow-up to results from random scheduled sampling). Then, the PHV performs a Kidney Inhibition Swab (KIS™) test (in-plant screening test) for the presence of antibiotic drug residues. If the KIS™ test result is positive, the sample is submitted to FSIS field laboratories for confirmation.

With this background information, we hypothesize that the use of injectable antimicrobial drugs may have increased in food animals in the U.S. after the implementation of change in VFD regulations, which could increase the detection of violative antibiotic residues in the tissue of food animals in the U.S. Violative antibiotic tissue residues may pose a risk of adverse health effects in humans, such as an increase in resistant bacteria [13,14], allergic reaction [14,15], altering gut microbiota [16] and obesity [16,17] from consuming such residues. To our knowledge, no study has quantified the association of the VFD rule changes on the presence of violative penicillin, tetracycline, sulfonamide, desfuroylceftiofur, florfenicol, and tilmicosin residues in the tissue of food animals in the US. These antibiotics are commonly used in food-producing animals in the U.S. Therefore, this study aimed to investigate the association of the implementation of revised VFD regulations on the detection of violative penicillin, tetracycline, sulfonamide, desfuroylceftiofur, tilmicosin, and florfenicol residues in the tissue of food animals from IGS samples in slaughterhouses in the U.S. Our study results could provide a baseline understanding of the relationship changes in VFD regulations to detection rates of violative residues of penicillin, tetracyclines, sulfonamides, desfuroylceftiofur, florfenicol, and tilmicosin in the tissue of food animals in the U.S.

## 2. Materials and Methods

### 2.1. Data Source

The inspector-generated sampling (IGS) data used for this study were retrieved from the U.S. NRP for meat, poultry, and egg products [18]. These data covered the period between 2014 and 2019. Penicillin, tetracyclines, sulfonamides, desfuroylceftiofur, florfenicol, and tilmicosin were selected as target antibiotics for analysis because they are commonly used in food animals in the U.S. and are important antibiotics in human health. 

### 2.2. Data Preparation and Variables

The IGS dataset contains the following variables: antibiotic residues (penicillin, tetracyclines, sulfonamides, desfuroylceftiofur, florfenicol, and tilmicosin), testing results (violative and non-violative), date of collection (month and year), animal species (cattle, goat, sheep, swine, and turkey), tissue name (kidney, liver, and muscle), and analyte name (drug name). The dataset was transferred from Microsoft Excel (version 2019, Microsoft Corporation, Redmond, WA, USA) to STAT 16.1 software (Stata Corporation, College Station, TX, USA). Then, all variables in the dataset were assessed for completeness and accuracy. Next, the ‘year’ variable was collapsed into a dichotomous variable, “VFD rule changes”: ‘after VFD rule changes (2017 to 2019) and ‘before VFD rule changes’ (2014 to 2016) for analysis. This VFD rule changes variable was the primary exposure of interest in this analysis. Others included the type of animal and type of tissue sampled. The animal species’ variable was considered the animal production class’ variable and was categorized based on production class such as bob veal, beef cow, dairy cow, bull, heifer, steer, goat, sheep, swine, and turkey. Besides, the ‘tissue name’ variable was collapsed into a dichotomous variable as ‘type of tissue sampled’ (kidney vs. others (liver/muscle)). Chlortetracycline, oxytetracycline, tetracycline, and doxycycline were aggregated as the antibiotic group ‘tetracyclines’. Similarly, sulfadiazine, sulfadimethoxine, sulfadoxine, sulfamethazine, and sulfamethoxazole were aggregated as the antibiotic group ‘sulfonamides’. The outcome of interest for each antibiotic or antibiotic group was whether violative residue was present compared to absence in the tissue of food animals from the IGS. 

### 2.3. Statistical Analyses

All statistical analyses were conducted using Stata 16.1 (Stata Corporation, College Station, TX, USA). Categorical predictor variables were summarized using frequencies and percentages. Chi-square or Fisher’s exact test (if the expected cell count was <5) was used to investigate the distribution of the outcome variables with respect to categorical predictor variables. The differences were then assessed for significance by *p*-values, with *p* < 0.05 considered significant. Separate logistic regression models were built for the seven antibiotic residues: penicillin, tetracyclines, sulfonamides, desfuroylceftiofur, tilmicosin, and florfenicol. 

Each model-building process involved two steps. The first step involved fitting univariable logistic regression models to assess crude associations between potential predictor variables and detection of violative residues in tissue samples. A relaxed α of 0.2 was used to identify potentially significant predictors, and variables with a *p* < 0.2 in the univariable analysis were considered for further investigation in multivariable models in step two. Pair-wise collinearity of these variables was examined in order to prevent the inclusion of collinear variables in the multivariable models. When two variables were highly correlated (absolute value of rho > 0.70; *p* < 0.05), only one was selected for consideration in the multivariable models. The decision regarding which of a pair of highly correlated variables to include in step two was based on biological and statistical considerations. 

The multivariable logistic regression model was initially built by fitting a full model that included all non-correlated variables with univariable *p* ≤ 0.20. In addition, the variable after VFD regulation rule changes was included in each full model regardless of the *p*-value obtained from univariable regression. Non-significant predictor variables were removed using manual backward elimination, with a critical *p*-value of ≤0.05. However, non-significant variables were considered potential confounders if their removal from the model resulted in a large (greater than 20%) change in the coefficients of any of the remaining variables in the model and were considered for retention in the final model. Two-way interaction terms between VFD rule changes, animal production class, and type of tissue sampled were assessed for statistical significance. The fitness of the final model was assessed using Hosmer-Lemeshow goodness-of-fit statistics [19]. When the Hosmer-Lemeshow goodness-of-fit test was not appropriate, the area under the curve (AUC) value was used to evaluate the final model. Results of the final model were reported as odds ratio (OR) with a 95% confidence interval (CI).

## 3. Results

The original IGS dataset contained 7762 records of testing results for drug residues in the tissues of food animals. A total of 4391 records contained results of testing for residues of the antibiotics of interest in this study (penicillin: 1310; tetracyclines: 983; sulfonamides: 901; desfuroylceftiofur: 809; florfenicol: 181; and tilmicosin: 207) and were included in the analysis.

### 3.1. Univariable Logistic Regression Results

Type of tissue samples was significantly associated with the detection of violative penicillin residue in the tissue of food animals from the IGS (Table 1). Similarly, animal production class and type of tissue sampled were significantly associated with the detection of violative tetracycline residues in the tissue of food animals from the IGS samples (Table 2). In addition, sample collection following the implementation of changes in VFD regulations was significantly associated with detecting violative sulfonamide residues in the tissue of food animals from the IGS (Table 3). Furthermore, the type of tissue sample was significantly associated with detecting violative desfuroylceftiofur residues in the tissue of food animals from the IGS (Appendix A). There was no statistically significant association between animal production classes, type of tissue sample, and VFD rule changes with the detection of violative tilmicosin (Appendix A) and florfenicol (Appendix A) residues in the tissue of food animals from the IGS. 

### 3.2. Multivariable Logistic Regression Results

In the final multivariable logistic regression model for penicillin, which included 1310 observations, significant predictors associated with detecting violative residues in the tissue of food animals included the type of tissue sampled (Table 4). The Hosmer-Lemeshow test was not used as a summary goodness-of-fit measure for the final penicillin model because there were only two covariate patterns (at least 6 covariate patterns should be present when using the Hosmer-Lemeshow test) [20]. Hence, the final penicillin model was assessed using the area under the curve (AUC), indicating the proportion of outcomes correctly classified by the model (AUC value = 0.5914).

The implementation of changes in VFD regulations was significantly associated with detecting violative penicillin residues in the tissue of food animals from the IGS. The odds of detecting penicillin residue violations decreased by 24% after the implementation of VFD regulations rule changes compared to before the VFD rule change implementation, and this finding was statistically significant (Table 4). 

The interaction term (VFD rule changes*type of tissue sampled) was statistically significant in the final model. Hence, we reported the relationship between types of tissue samples and detecting violative penicillin residues in the tissue of food animals by VFD rule change categories (before VFD rule change and after VFD rule change). The odds of detecting penicillin residue violations was about 4 times higher in the kidney than in other tissue (muscle) before implementing the VFD rule change (Table 5). However, the odds of detecting penicillin residue violations was about 13 times higher in kidneys than in other tissue after implementing the VFD rule change (Table 5). 

The final multivariable logistic regression model for tetracycline had 960 observations. The type of animal and type of tissue sampled were significant predictors of tetracycline residue violations in food animal tissues from the IGS (Table 6). The p-value for the Hosmer-Lemeshow test was 0.0833, indicating that the final tetracycline model fit the data well. 

Animal production class was significantly associated with detecting violative tetracycline residues in the tissue of food animals. The magnitude of association varied according to animal production class. For example, the odds of detecting violative tetracycline residues in the tissue of bob veal was 74% decreased compared to the tissue of dairy cows (Table 6). On the other hand, the odds of detecting violative tetracycline residues in the tissue of sheep was 40 times higher than in the tissue of dairy cows (Table 6). The odds of detecting violative tetracycline residues were about 8 times high in other tissue (muscle) samples compared to kidney samples (Table 6). 

Although the odds of detecting violative tetracycline residues were 54% higher for samples collected following the implementation of the VFD rule change compared to those collected prior to the VFD rule change, this finding was not statistically significant (Table 6). Again, none of the interaction terms assessed (VFD rule changes*type of animal and VFD rule changes*type of tissue sampled) were statistically significant in the final tetracycline model.

The final multivariable logistic regression model for sulfonamides had 901 observations (Table 7). The Hosmer-Lemeshow test was not used as a summary goodness-of-fit measure for the final sulfonamide model because there were only two covariate patterns (at least 6 covariate patterns should be present when using the Hosmer-Lemeshow test) [20]. Hence, the final sulfonamide model was assessed using the area under the curve (AUC), indicating the proportion of outcomes correctly classified by the model (AUC value = 0.56). The implementation of changes in VFD regulations was significantly associated with detecting violative sulfonamide residues in the tissues of food animals. The odds of detecting sulfonamide residue violations decreased by 36% after the implementation of changes in VFD regulations compared to before the VFD rule change period, and this finding was statistically significant (Table 7). 

Regarding desfuroylceftiofur, the final multivariable logistic regression model had 809 observations (Appendix A). The final model was assessed using the area under the curve (AUC), indicating the proportion of outcomes correctly classified by the model (AUC value = 0.5308). Although the odds of detecting violative desfuroylceftiofur residues were 2% decreased for samples collected following the implementation of the VFD rule change compared to those collected before the VFD rule change, this finding was not statistically significant (Appendix A). The odds of detecting desfuroylceftiofur residue violations was 12 times higher in the kidney than in other tissue (muscle) (Appendix A). 

The final multivariable logistic regression model for tilmicosin had 207 observations (Appendix A). The final model was assessed using the area under the curve (AUC), indicating the proportion of outcomes correctly classified by the model (AUC value = 0.5124). The odds of detecting violative tilmicosin residues were 10% decreased for samples collected following the implementation of the VFD rule change compared to those collected before the VFD rule change. However, this finding was not statistically significant (Appendix A). 

Regarding florfenicol, the final multivariable logistic regression model had 181 observations (Appendix A). The final model was assessed using the area under the curve (AUC), indicating the proportion of outcomes correctly classified by the model (AUC value = 0.5407). The odds of detecting violative florfenicol residues were 28% decreased for samples collected following the implementation of the VFD rule change compared to those collected before the VFD rule change. However, this finding was not statistically significant (Appendix A).

## 4. Discussion

To the best of our knowledge, this is the first report describing the association of changes in VFD regulations on the detection rates of violative penicillin, tetracycline, sulfonamide, desfuroylceftiofur, tilmicosin, and florfenicol residues in the tissues of food animals in slaughterhouses in the U.S. Our study highlights three critical findings. Firstly, compared to the period before changes in VFD regulations, the odds of detecting violative sulfonamide and penicillin residues in the tissues of food animals sampled (from the IGS) following VFD implementation decreased by 36% and 24%, respectively, irrespective of the animal production class. Secondly, animal production class was significantly associated with the detection of violative tetracycline residues. However, the implementation of change in the VFD rule was not significantly associated with the tetracycline residue violation rates in the tissue of food animals from the IGS. Finally, the type of tissue sampled was significantly associated with tetracycline and desfuroylceftiofur residues violation. However, the implementation of the change in VFD rule was not associated with the desfuroylceftiofur residues violation rates in the tissue of food animals from the IGS.

Before this study, cattle producers perceived that changes in VFD regulations would lead to increased use of injectable antibiotics by producers [8] and an overall increase in residue violations. Results of the current study showed that after the implementation of change in the VFD rule, the detection of violative sulfonamide and penicillin residues decreased significantly in the tissue of food animals from the IGS. There are several potential explanations for these findings. For instance, revised VFD regulations may not have impacted the use of sulfonamide and penicillin injectables. Alternatively, the use of injectable sulfonamides and penicillin may have increased following the implementation of VFD regulations; however, the relatively short withdrawal period (Sulfonamides:5 days; penicillin G: range from 4 to 10 days, as label withdrawal time) [21] may have increased the likelihood of farmers’ compliance, leading to non-violative residues in our study. Other potential factors could be associated with this finding depending on dose/route/duration and animal production class. Payne MA et al., [22] reported that extra-label use of penicillin in food-producing animals under the direction of a veterinarian as the labeled dose of penicillin is not effective, and the extra-labeled requires an extended withdrawal period, typically at least 21–30 days depending on dose/route/duration [22]. Also, clinical illness can impact the withdrawal time (as the withdrawal time is established in healthy animals), and may also play role in the risk of antibiotic residue violation in tissues of food animals.

In contrast, the odds of detecting violative tetracycline residues among samples collected following the implementation of change in the VFD rule were not decreased significantly compared to before the implementation of change in the VFD rule. Multiple factors could explain these findings. Farmers have expressed displeasure with rule changes in VFD regulations because non-therapeutic use of medically important antimicrobials in medicated feed for growth promotion and feed efficiency, which was, permitted prior to implementation of VFD rule changes, may have prevented or reduced clinical diseases later in animals’ life [8]. If cases of clinical disease among food animals were more frequent following changes in VFD regulations, injectable (including extra-label) use of tetracyclines might have increased to treat these animals. Furthermore, injectable tetracycline has relatively lengthy withdrawal periods of 28 days [21]. Adhering to these withdrawal periods could be more challenging than sulfonamides, leading farmers to send treated animals to slaughter with violative tissue levels of antibiotic residues. In addition, farmers with limited experience using injectable antibiotics may be unaware of proper dosing. Hence, imprudent use of tetracycline, including incorrect dosage and route of administration [23,24,25], may have contributed to the residue violations observed in this study. Previous studies have reported that failure to follow meat withdrawal periods and extra-label use of injectable tetracycline may be associated with antibiotic residues in the tissues of food animals [23,26,27,28]. Future studies are warranted to investigate practices of injectable antibiotic administration, including extra-label use, treatment documentation, and knowledge of antibiotic withdrawal periods in food animals with clinical illnesses, to elucidate the spectrum of these issues (after the implementation of changes in VFD regulations) at the farm level in the U.S. 

This study revealed significant differences in the odds of detecting violative tetracycline and penicillin residues between kidney and other tissue samples (muscle/liver). Kidney tissue samples had higher odds of penicillin residue violations than samples from other tissues (muscle/liver). This magnitude of association varied before and after VFD rule changes; for instance, higher odds (OR = 13.14) of detecting violative penicillin residues after VFD rule change than before the implementation of VFD rule changes (OR = 3.95). This is an expected finding because most (60–90%) of parenterally administered penicillin is eliminated in the urine, and kidneys represented the majority (92%) of the sampled tissues in the dataset. Hence, this finding is consistent with the results of Paturkar et al. [29], regardless of any regulatory change.

On the other hand, muscle tissue samples had higher odds of tetracycline residue violations compared to kidney samples. Several factors may have contributed to this finding, including the route of administration and extra-label use of tetracycline in food animals. For example, tetracycline residues have been found at the injection site as many as 35 days after intramuscular administration [30]. In addition, previous studies have reported that tetracycline residue levels were higher in muscles than in kidneys [31,32,33,34], regardless of regulations. Future experimental and epidemiologic field studies could generate knowledge on host- and farm-level factors associated with tissue levels of tetracycline residues in food animals in the U.S. 

The results of multivariable logistic regression models showed that bob veal samples had lower odds of residue violations for tetracycline compared to dairy cows. On the other hand, compared to dairy cows, sheep and goats had higher odds of detecting tetracycline residue violations. This finding indicates that withdrawal times set for antibiotic use in goats and sheep are not always followed or are inaccurate because the use of antibiotics in goats and sheep is predominantly extra-label [35,36]. Practices of extra-label antibiotics could be more common in sheep and goats [30] because there are limited FDA-approved labeled antibiotic products in the U.S. [37] A study reported that extra-label antibiotic use is more common in small ruminants than in cattle [37]. This inappropriate or extra-label antibiotic use in these animal classes [38] may play a role in the risk of tetracycline residue violations. However, extra-label use of medicated feed is not prohibited in these animal classes [37]. It requires a written recommendation by a licensed veterinarian within the confines of a valid veterinarian-client-patient relationship in the U.S [37]. A previous study reported that goats had a higher frequency of antibiotic residues at slaughter in Missouri [38]. A study from Alberta indicates that tetracycline was one of the most common injectable antibiotics used in sheep [39]. Our findings warranted further research and highlighted that increased labeled antibiotic options for these animal classes would provide producers with appropriate withdrawal times to follow. Also, there is a need to improve working relationships between veterinarians and goat/sheep farmers’ to promote appropriate antibiotic use [38] to prevent the occurrence of antibiotic residues in the tissue of these animal classes at slaughter.

The result of this study would not be generalizable because the tissue samples were collected using the targeted sampling of food-producing animals under the IGS. The tissue samples from the IGS were chosen based on clinical signs or pathologic lesions on food-producing animals during antemortem and post-mortem examination by a veterinarian authorized to collect the tissue samples. So current study results may over-represent the violation of antibiotic residues in tissues of food-producing animals from the IGS than all other food-producing animals brought to the slaughterhouse. Our study findings only apply to the samples collected under the IGS, not the entire food-producing animals brought into the slaughterhouse.

Besides, our study used at least 181 observations for each class of antibiotic of interest, and a larger sample size could be more helpful. However, given the number of covariates used in our model, we consider this sample size adequate for the study. In addition, there were limited variables in the dataset, so we suggest including animal-level information such as age, sex, breed, pathologic lesions or signs, and location (state-level) of sampled animals under the IGS scheme. Besides, results of the animal production class should be generalized cautiously because dairy cows are used as the reference population (as there is no VFD use of antimicrobials in dairy cattle) in the animal production class variable for all analyses in the study. Cull dairy cows are one of the most likely production categories to have violative residues identified, although this varies based on antibiotic class. For example, penicillin was the most frequently identified antibiotic with violative residue levels in culled cows in the U.S. [5].

## 5. Conclusions

In summary, the implementation of the VFD rule changes in 2017 did not increase the detection of violative residues of injectable antibiotics in the tissues of food animals from the IGS. Actually, the VFD rule changes had a positive impact on violative residues of a few injectable antibiotics. Violative residues of sulfonamides and penicillin were reduced, but violative residues of tetracyclines, desfuroylceftiofur, tilmicosin, and florfenicol did not change. In addition to the practical benefits of the VFD rule changes, multi-sectoral coordinated educational interventions to food animal producers and farmers concerning withdrawal periods, record-keeping, and compliance with label instructions of antibiotics is critical. Such wholistic approach would further reduce violative antibiotic residues in the tissues of food animals in the U.S.

## Figures and Tables

**Table 1 microorganisms-10-02031-t001:** Univariable association between predictors and detection of violative residues of penicillin in the tissue of food animals (*n* = 1310) from the IGS, 2014–2019.

Predictor	Categories	Violation N (%)	Non-Violation N (%)	OR	95% CI	*p*-Value
VFD rule change						0.116
	Before VFD rule change (2014–2016)	460 (72)	182 (28)	Referent		
	After VFD rule change (2017–2019)	452 (68)	216 (32)	0.82	0.65, 1.04	0.117
Animal production class						0.501
	Bob veal	58 (65)	31 (35)	0.91	0.57, 1.45	0.704
	Beef cow	74 (73)	27 (27)	1.33	0.83, 2.14	0.225
	Dairy cow	430 (67)	210 (33)	Referent		
	Bull	117 (71)	47 (29)	1.21	0.83, 1.77	0.309
	Heifer	131 (64)	75 (36)	0.85	0.61, 1.18	0.343
	Steer	17 (74)	6 (26)	1.38	0.53, 3.56	0.501
	Goat	2 (50)	2 (50)	0.48	0.06, 3.49	0.475
	Sheep	4 (100)	0 (0)	1	NA	NA
	Swine	36 (100)	0 (0)	1	NA	NA
	Turkey	43 (100)	0 (0)	1	NA	NA
Type of tissue sampled						<0.001
	Muscle	34 (32)	73 (68)	Referent		
	Kidney	878 (73)	325 (27)	5.8	3.78, 8.88	<0.001

95% confidence interval (CI); odds ratio (OR); NA (not applicable).

**Table 2 microorganisms-10-02031-t002:** Univariable association between predictors and detection of violative residues of tetracycline in the tissue of food animals (*n* = 983) from the IGS, 2014–2019.

Predictor	Categories	Violation N (%)	Non-Violation N (%)	OR	95% CI	*p*-Value
VFD rule change						0.244
	Before VFD rule change (2014–2016)	40 (8)	465 (92)	Referent		
	After VFD rule change (2017–2019)	48 (10)	430 (90)	1.29	0.83, 2.01	0.245
Animal production class						<0.001
	Bob veal	13 (5)	267 (95)	0.45	0.22, 0.90	0.024
	Beef cow	17 (10)	150 (90)	1.05	0.55, 2.00	0.863
	Dairy cow	27 (10)	252 (90)	Referent		
	Bull	10 (12)	75 (88)	1.24	0.57, 2.68	0.578
	Heifer	7 (7)	96 (93)	0.68	0.28, 1.61	0.383
	Steer	1 (5)	19 (95)	0.49	0.06, 3.81	0.497
	Goat	8 (40)	12 (60)	6.22	2.33, 16.55	<0.001
	Sheep	5 (83)	1 (17)	46.66	5.25, 414.23	0.001
	Swine	0 (0)	12 (100)	1	NA	NA
	Turkey	0 (0)	11 (100)	1	NA	NA
Type of tissue sampled						0.002
	Kidney	80 (8)	882 (92)	Referent		
	Others (muscle)	8 (38)	13 (62)	6.78	2.73, 16.85	<0.001

95% confidence interval (CI); odds ratio (OR); NA (not applicable).

**Table 3 microorganisms-10-02031-t003:** Univariable association between predictors and detection of violative residues of sulfonamides in the tissue of food animals (*n* = 901) from the IGS, 2014–2019.

Predictor	Categories	Violation N (%)	Non-Violation N (%)	OR	95% CI	*p*-Value
VFD rule change						0.014
	Before VFD rule change (2014–2016)	417 (87)	64 (13)	Referent		
	After VFD rule change (2017–2019)	339 (81)	81 (19)	0.64	0.44, 0.91	0.015
Animal production class						0.082
	Bob veal	188 (91)	19 (9)	2.01	1.16, 3.48	0.012
	Beef cow	42 (88)	6 (12)	1.42	0.57, 3.50	0.441
	Dairy cow	290 (83)	59 (17)	Referent		
	Bull	70 (79)	19 (21)	0.74	0.42, 1.33	0.329
	Heifer	100 (78)	28 (22)	0.72	0.43, 1.20	0.214
	Steer	33 (83)	7 (17)	0.95	0.40, 2.27	0.924
	Goat	7 (78)	2 (22)	0.71	0.14, 3.51	0.677
	Sheep	1 (100)	0 (0)	1		
	Swine	20 (83)	4 (17)	1.01	0.33, 3.08	0.976
	Turkey	5 (83)	1 (17)	1.01	0.11, 8.86	0.988
Type of tissue sampled						NA
	Others (muscle/liver)	642 (82)	145 (18)	Referent		
	Kidney	114 (100)	0 (0)	1	NA	NA

95% confidence interval (CI); odds ratio (OR); NA (not applicable).

**Table 4 microorganisms-10-02031-t004:** Results of multivariable logistic regression for predictors of detection of violative residues of penicillin in the tissue of food animals (*n* = 1310) from the IGS, 2014–2019.

Predictor	Categories	OR	95% CI	*p*-Value
VFD rule change				0.030
	Before VFD rule change (2014–2016)	Referent		
	After VFD rule change (2017–2019)	0.76	0.59, 0.97	0.031
Type of tissue sampled				<0.001
	Others (muscle)	Referent		
	Kidney	6.01	3.91, 9.23	<0.001
VFD rule change*type of tissue sampled		0.3009283	0.11, 0.80	0.017

95% confidence interval (CI); odds ratio (OR); interaction (*) between VFD rule change and type of tissue sampled.

**Table 5 microorganisms-10-02031-t005:** Results of association between type of tissue sampled and penicillin residues in the tissue of food animals by VFD rule change categories.

Predictor	Categories	OR	95% CI	*p*-Value
Before the VFD rule change (2014–2016), *n* = 642				
Type of tissue sampled	Others (muscle)	Referent		
	Kidney	3.95	2.32, 6.73	<0.001
After the VFD rule change (2017–2019), *n* = 668				
Type of tissue sampled	Others (muscle)	Referent		
	Kidney	13.14	5.75, 30.02	<0.001

95% confidence interval (CI); odds ratio (OR).

**Table 6 microorganisms-10-02031-t006:** Results of multivariable logistic regression for predictors of detection of violative residues of tetracyclines in the tissue of food animals (*n* = 960) from the IGS, 2014–2019.

Predictor	Categories	OR	95% CI	*p*-Value
VFD rule change				0.092
	Before VFD rule change (2014–2016)	Referent		
	After VFD rule change (2017–2019)	1.54	0.93, 2.55	0.092
Animal production class				0.001
	Dairy cow	Referent		
	Bob veal	0.36	0.17, 0.76	0.007
	Beef-cow	0.97	0.50, 1.88	0.942
	Bull	0.98	0.43, 2.20	0.962
	Heifer	0.56	0.22, 1.39	0.218
	Steer	0.54	0.06, 4.24	0.562
	Goat	6.11	2.27, 16.47	<0.001
	Sheep	40.24	4.45, 363.69	0.001
	Swine	1		
	Turkey	1		
Type of tissue sampled				<0.001
	Kidney	Referent		
	Others (muscle)	7.71	3.02, 19.70	<0.001

95% confidence interval (CI); odds ratio (OR).

**Table 7 microorganisms-10-02031-t007:** Results of multivariable logistic regression for predictors of detection of violative residues of sulfonamides in the tissue of food animals (*n* = 901) from the IGS, 2014–2019.

Predictor	Categories	OR	95% CI	*p*-Value
VFD rule change				0.014
	Before VFD rule change (2014–2016)	Referent		
	After VFD rule change (2017–2019)	0.64	0.44, 0.91	0.015

95% confidence interval (CI); odds ratio (OR).

## Data Availability

All the datasets used in this study are secondary datasets, and the links to access the datasets are given in this article. These data sets are publicly available, and no specific rights are required to access these datasets.

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
