# Peer review of "Effect of Changes in Veterinary Feed Directive Regulations on Violative Antibiotic Residues in the Tissue of Food Animals from the Inspector-Generated Sampling in the United States"

_microorganisms, 2022, doi:10.3390/microorganisms10102031_

Round 1

Reviewer 1 Report

The work deals about the relevant theme of antibioti residue occurrance in animal tissues. A very interesting perspective has been applied, a focus on  consequences derived from a specific regulation, the Veterinary Food Directive. Studying the effects of a technical legislation on professional practice and behaviours in the field of antibiotic use can help in increasing safety

Author Response

Thank you so much, for your valuable review and comments.

Reviewer 2 Report

It is very important that study the effect of VFD on the phenomenon of excessive antibiotics in animals, which can guide the establishment of various regulations. In this paper, the analysis of various data is very thorough, and the only disadvantage is that the sample size is smaller.

Author Response

Thank you so much for your valuable review and comments. In this study, we analyzed data for specific antibiotics such as penicillin (n=1310), tetracycline (n=983), and sulfonamides (n=901) that meet the antibiotics of interest in this study. We stated this information in the manuscript's result section (yellow color highlighted). We had originally thought that the sample size was not small enough to acknowledge as a major limitation in our discussion. However, after your comment, we have included a statement addressing the sample size in our last discussion paragraph in the revised manuscript (yellow color highlighted). The statement is "Our study used at least 900 observations for each class of antibiotics of interest. Higher sample size could be more helpful. However, given the number of covariates used in our model, we consider this sample size adequate for the study".

Author Response

Thank you so much for your suggestions. We totally agree with you that the maximum residue levels of tetracycline are very high for kidneys i.e. 1200 μg/kg in kidney while 200 μg/kg in the muscle (FAO/WHO). In our study, we examined whether the violative level (exceeded the MRL) of tetracycline residues varied significantly by type of tissue (kidney vs. muscle) using the logistic regression model. Besides, we also discussed the possible explanation why muscle tissue had a higher odds of the violative level of tetracycline residues compared to kidneys (yellow color highlighted)) in the discussion section of the manuscript. We checked our revised manuscript by a native English-speaking colleague to improve the English language and style.  In the revised draft of the manuscript, all track changes to show that a native English speaker reviewed the article as suggested.

Reviewer 4 Report

Th manuscript that deals with antimicrobial residues in tissues of food animals befor and after the U.S. Veterinary Feed Directive (VFD) is important for the U.S. and all countries that have not started any national monitoring and surveillance programmes to curb the overuse of antibiotics to minimize antimicrobial resistance - in the European Union (EU), the focus on the antibiotic use in food animals has shifted from reducing residues to minimizing antimicrobial resistance, since residues in animal tissues are not any longer seen as teh major problem of antimicrobial use. However, to study, whether the VFD has a measurable effect on the frequency of violative residues is very important for countries that plan to regulate the judicious use of antibiotics.

The result that the frequency of violative residues sulfonamides was significantly reduced after the VFD was impelmented, but not the frequencies of violative residues of penicciline and tertracycline is very interesting and the conclusion that this effect should be further investigated, is highly justified.

In line 73, the sentence: "This study results could..." should start with "These study results.." However, apart from this, the English is excellent.

In Line 97, the abbreviation "IGS data" is nowhere spelled out und should be explained.

Despite the accurate English, the understandability coulde in some cases be improved: the wording of statisticians need to be adapted to the addressed readers such as veteerinarians and regulators.

Here is an example (line 218/219): "...the type of animal and type of tissue sampled were significantly associated with residue violation, but VFD regulation was not." The reviewer asks, how a regulation can be associated with something? The reviewer guesses that the impact of the VFD is meant and not the regulation itself?? This should be put in better words. This wording should also clarified, wherever else it is used in the text, and in the summary!!

Author Response

Thank you so much, for your comments and suggestions.  We have changed “This” to “these…”  (yellow color highlighted) in the revised draft of the manuscript.

We elaborated the IGS in the revised draft of the manuscript (yellow color highlighted). Also, we described the IGS in the introduction section of the revised manuscript (yellow color highlighted).

We revised the wording of the ‘VFD regulation’ to ‘VFD regulation implementation period did not impact on …….. residue violation’ in the discussion section of the revised draft of the manuscript (yellow color highlighted). Besides, we also revised the word in the summary section of the revised manuscript.

Reviewer 5 Report

September 1, 2022 -- Reviewer Comments:

(A Word document of the manuscript is attached with the editing changes made by this reviewer highlighted in green.)

A citation is needed for the reference to the FDA Guidance for Industry #213 in the Introduction.

Throughout the attached Word document file, changes were made to correctly reflect the situation regarding FDA’s changes to the Veterinary Feed Directive rules which were made in October 1, 2015, and were fully implemented beginning on January 1, 2017.    This January 1st date in 2017 was the point in time when drug manufacturers, veterinarians, and livestock owners had to comply with these major rule changes for obtaining and using VFD drugs in medicated feed.  As initially written, the authors seemingly present a misconception in this manuscript that VFD regulations were initiated beginning in 2017.  The initial VFD rule was developed in 1996 by the FDA, not in 2017.  The authors need to carefully go through all parts of the manuscript to ensure that the needed edits are made to reflect that changes were made in the VFD rules, effective on January 1, 2017, instead of stating or implying that VFD regulations began in 2017 (implying that VFD regulations did not exist before this time).

Within the introduction and continuing throughout the manuscript is a lack of mentioning the main reason why FDA undertook the process to change the VFD regulations.  As clearly stated in GFI #213, “Specifically, the final guidance discusses FDA’s concerns regarding the development of antimicrobial resistance in human and animal bacterial pathogens when medically important antimicrobial drugs are used in food-producing animals in an injudicious manner.” Incorporation of this point into the manuscript would seem to be important for giving the reader better understanding about the motivation by FDA to make these major changes in the VFD rules. 

Concerns in the manuscript related to illegal extralabel drug use (ELDU) of drugs administered in feed

Since the adoption of Animal Medicinal Drug Use Clarification Act (AMDUCA) regulations in 1994, ELDU of any drug, including antimicrobials, is prohibited. 

Lines ~ 289 – 292 (showing recommended edits by reviewer in green highlighting): 

Farmers have expressed displeasure with rule changes in VFD regulations because non-therapeutic use of medically important antimicrobials in medicated feed for growth promotion and feed efficiency, which was permitted prior to implementation of VFD rule changes, may have pre-vented or reduced clinical diseases later in animals’ life [7].

The authors should understand that owners were not legally permitted to use antimicrobials in medicated feed for any therapeutic purpose, including disease prevention, unless that disease indication (i.e., prophylaxis or disease prevention) was listed on the label as an approved indication.  Prior to the VFD final rule change in 2017, owners were permitted to use medically important antimicrobials in feed for growth promotion and feed efficiency (non-therapeutic uses) and for therapeutic purposes (i.e., prevention, control, or treatment) provided that any intended uses of medication in feeds had the appropriate indications on the labels.  The owners were required by these AMDUCA regulations to explicitly follow the label directions for concentration of drugs in the feed, for dosages of the drugs (based on feed intake), and any other requirements or instructions listed as part of the label directions.  In other words, ELDU in feed has always been prohibited by FDA for feeding of medicated feed to groups of animals.  That prohibition still continues after the adoption of the VFD final rule change.  (Prior to AMDUCA, all ELDU of all animal drugs were technically illegal, and FDA generally followed a policy of discretionary enforcement until AMDUCA regulations were enacted.)

Lines ~333-335:

Furthermore, it is difficult for livestock farmers who practice extra-label antibiotic use to comply with the recommended withdrawal period or determine the appropriate dose for each animal.

This statement is very troubling.  It is clearly illegal for livestock farmers to “practice extra-label antibiotic use.”  AMDUCA rules are very clear that ELDU is permitted only by or under the supervision of a licensed veterinarian with a VCPR.  Veterinary supervision and oversight are required because licensed veterinarians have the appropriate scientific and medical knowledge, training, and skills to properly establish extended withdrawal intervals when ELDU is involved.  Livestock owners are not recognized by FDA to be qualified or legally entitled to determine extended withdrawal intervals on their own.  Similarly, livestock owners are not recognized by FDA to be qualified or legally entitled to modify the dosages for each animal.  A licensed veterinarian with a VCPR must be involved in ELDU if it is to be medically legal, according to the regulations.  

This reviewer strongly believes that the microorganisms journal should not allow the publication of information in its articles that would be viewed by FDA as endorsing illegal activity being practiced by owners.

Author Response

Comments and Suggestions for Authors

 (A Word document of the manuscript is attached with the editing changes made by this reviewer highlighted in green.)

Response: Thank you so much for your thorough review and suggestions, which will be improved the revised version of the manuscript. 

A citation is needed for the reference to the FDA Guidance for Industry #213 in the Introduction.

Response: The citation has been included in the revised version of the manuscript.

Throughout the attached Word document file, changes were made to correctly reflect the situation regarding FDA’s changes to the Veterinary Feed Directive rules which were made in October 1, 2015, and were fully implemented beginning on January 1, 2017.    This January 1st date in 2017 was the point in time when drug manufacturers, veterinarians, and livestock owners had to comply with these major rule changes for obtaining and using VFD drugs in medicated feed.  As initially written, the authors seemingly present a misconception in this manuscript that VFD regulations were initiated beginning in 2017.  The initial VFD rule was developed in 1996 by the FDA, not in 2017.  The authors need to carefully go through all parts of the manuscript to ensure that the needed edits are made to reflect that changes were made in the VFD rules, effective on January 1, 2017, instead of stating or implying that VFD regulations began in 2017 (implying that VFD regulations did not exist before this time).

Response: Thank you so much for your comments, suggestions, and editing changes. We have corrected the VDF date error in the revised version of the manuscript. Besides, we have changed the VFD regulation to the VFD rule change in the revised version of the manuscript.

Within the introduction and continuing throughout the manuscript is a lack of mentioning the main reason why FDA undertook the process to change the VFD regulations.  As clearly stated in GFI #213, “Specifically, the final guidance discusses FDA’s concerns regarding the development of antimicrobial resistance in human and animal bacterial pathogens when medically important antimicrobial drugs are used in food-producing animals in an injudicious manner.” Incorporation of this point into the manuscript would seem to be important for giving the reader better understanding about the motivation by FDA to make these major changes in the VFD rules. 

Response: Thank you so much for pointing out this issue and suggestion. We have incorporated the main reason why FDA undertook the process to change the VFD regulations in the revised version of the manuscript (Line 38-50).

Concerns in the manuscript related to illegal extralabel drug use (ELDU) of drugs administered in feed. Since the adoption of Animal Medicinal Drug Use Clarification Act (AMDUCA) regulations in 1994, ELDU of any drug, including antimicrobials, is prohibited. Lines ~ 289 – 292 (showing recommended edits by reviewer in green highlighting): 

Farmers have expressed displeasure with rule changes in VFD regulations because non-therapeutic use of medically important antimicrobials in medicated feed for growth promotion and feed efficiency, which was permitted prior to implementation of VFD rule changes, may have pre-vented or reduced clinical diseases later in animals’ life [7].

Response: Thank you so much for the edits. We have incorporated these edits in the revised version of the manuscript (Line 373-378).

The authors should understand that owners were not legally permitted to use antimicrobials in medicated feed for any therapeutic purpose, including disease prevention, unless that disease indication (i.e., prophylaxis or disease prevention) was listed on the label as an approved indication.  Prior to the VFD final rule change in 2017, owners were permitted to use medically important antimicrobials in feed for growth promotion and feed efficiency (non-therapeutic uses) and for therapeutic purposes (i.e., prevention, control, or treatment) provided that any intended uses of medication in feeds had the appropriate indications on the labels.  The owners were required by these AMDUCA regulations to explicitly follow the label directions for concentration of drugs in the feed, for dosages of the drugs (based on feed intake), and any other requirements or instructions listed as part of the label directions.  In other words, ELDU in feed has always been prohibited by FDA for feeding of medicated feed to groups of animals.  That prohibition still continues after the adoption of the VFD final rule change.  (Prior to AMDUCA, all ELDU of all animal drugs were technically illegal, and FDA generally followed a policy of discretionary enforcement until AMDUCA regulations were enacted.)

Lines ~333-335:Furthermore, it is difficult for livestock farmers who practice extra-label antibiotic use to comply with the recommended withdrawal period or determine the appropriate dose for each animal.

 This statement is very troubling.  It is clearly illegal for livestock farmers to “practice extra-label antibiotic use.”  AMDUCA rules are very clear that ELDU is permitted only by or under the supervision of a licensed veterinarian with a VCPR.  Veterinary supervision and oversight are required because licensed veterinarians have the appropriate scientific and medical knowledge, training, and skills to properly establish extended withdrawal intervals when ELDU is involved.  Livestock owners are not recognized by FDA to be qualified or legally entitled to determine extended withdrawal intervals on their own.  Similarly, livestock owners are not recognized by FDA to be qualified or legally entitled to modify the dosages for each animal.  A licensed veterinarian with a VCPR must be involved in ELDU if it is to be medically legal, according to the regulations.  

This reviewer strongly believes that the microorganisms journal should not allow the publication of information in its articles that would be viewed by FDA as endorsing illegal activity being practiced by owners.

Response: Thank you so much for your valuable comments and suggestions. We have removed the statement “Furthermore, it is difficult for livestock farmers who practice extra-label antibiotic use to comply with the recommended withdrawal period or determine the appropriate dose for each animal” from the revised version of the manuscript.

Reviewer 6 Report

Introduction:

Line 29-30:  This statement is unnecessarily inflammatory.  There are many other potential reasons for potential antibiotic residues in food products that aren’t “irrational”.  Strongly recommend “Inappropriate us of antibiotics in food animals is one factor associated with the presence of violative antibiotic residues in food animal products.”  This same sentiment is repeated in lines 44-45, so I would recommend removing there so as not to repeat.

Line 58 – no need to repeat the (IGS) abbreviation here.

Line 75-77 – this study does NOT add any information regarding whether violative levels of antibiotics are increasing or decreasing in tissues of food animals sold in grocery stores as this was not the sample tested.  The second half of the sentence after the comma referencing that should be removed.

M&M:

Lines 89-106 – The methodology used to collapse the data is extremely flawed.  The collapsing of all cattle data into a single variable, and all other species into a separate variable demonstrates a lack of understanding of these production systems and how the use of the VFD, types/routes of antibiotics frequently used, and differences in the levels of violative residues for the antibiotics tested differs between these species.  The cattle data should have been analyzed based on production class, which is given in the dataset (dairy cow, bull, beef cow, bob veal, heifer).  For example, there is no VFD use of antimicrobials in dairy cattle, therefore, there should be no change in antibiotic use related the implementation of the VFD in this class. Antibiotic use in bob veal calves, beef cattle, and swine, however, was likely to experience significant changes after VFD implementation.  In addition, collapsing all of the other animal species into a single variable does not take into account that for some animal species there is a tolerance set, and for others that don’t have labeled antibiotics (ie – goats) there is no tolerance so any residue is a violative residue.  At minimum these should be analyzed separately based on this information.

General comment – if the hypothesis was as stated “…that the use of injectable antimicrobial drugs may have increased in food animals in the U.S. after the implementation of the VFD, which could increase the detection of violative levels of residues in tissues of food animals in the U.S.”, why were residues for the most common classes of injectable antimicrobials not analyzed (ie – desfuroylceftiofur in dairy cattle; florfenicol, tilmicosin, tulathromycin in beef cattle, etc)?  This suggests a lack of understanding of production practices and antibiotic use in food producing animals in the US, as penicillin, oxytetracycline and sulphonamides are not the most commonly used injectable products.

Results:

Tables 1-3 – why was cattle used as referent for first two, and other used as referent for the other?

Discussion:

Line 267-269 – this sentence is not complete, needs edited. “Secondly, while animal and tissue type were significantly associated with penicillin residue violation, VFD implementation did not impact penicillin residue violation.”  Also need to edit 271-273.

Line 277-279 – The second sentence overreaches in regards to the the results of this study.  As the authors only looked at a small number of antibiotics, primarily older classes of drugs that are not necessarily the most commonly used antimicrobials on farms, they cannot holistically state that the VFD did or did not result in an increased used of injectable antibiotics.  Recommend removal of second sentence. 

Paragraph 287-208 – All veterinary-directed penicillin use in cattle is extralabel as the labeled dose is not effective.  This is not currently discussed but is an important part of the discussion that should be included, particularly as the original label on the bottle which is available over the counter currently lists a much shorter withdrawal. This discrepancy likely leads to errors in withdrawal time.  All of the antibiotics selected for study in this paper are currently available as OTC antibiotics, which may play a key role in the risk of residues, yet this is not discussed anywhere in the paper – this needs to be added.   Clinical illness can also impact the withdrawal time (as these are established in healthy animals), and may also play a role in these violative residues, and was not discussed

Paragraph 325-335 – This entire paragraph is very problematic.  Was the level required for violative residue considered in the model?  If no label, then any residue = violative.  Minor animal species have very few labeled drugs (ie – goats only have ceftiofur, sheep only ceftiofur, tilmicosin, and oxytetracycline).  See comment from M&M above - I do not think it was appropriate to lump all of these non-cattle species together for the analysis because of this issue.  Statement 328-330 is incorrect as written and should be removed or reworded.  Extralabel use in these species (ie - sheep and goats) is common simply because there is no labeled products, so it is a necessity that is perfectly legal.  As currently written, the results of this study should be a call for increased labeled antibiotic options for these animals which would then provide producers with appropriate withdrawal times to follow and establish tolerance levels.  333-334 is also an incorrect statement as extralabel drug use is perfectly legal if done under a valid VCPR, and the veterinarian of record is responsible for establishing a withdrawal.  Please reword to reflect this, or remove.

Overall recommendations:

While the study seeks to address an important question regarding antimicrobial use and residues following implementation of the VFD, the current data analysis has significant flaws as stated above, and is not publishable in its current state because of these flaws.  To address these issues, the data analysis could be re-run taking into consideration the issues identified above (preferred method).  The authors could instead attempt to clearly articulate the flaws in their study design in the discussion section, however, this is not the preferred method and may not be sufficient to address concerns identified.  It is also recommended that the authors seek out additional expertise in pharmacology and production animal medicine to assist them in their analyses given the concerns that were identified.

Author Response

Comments and Suggestions for Authors

Introduction:

Line 29-30:  This statement is unnecessarily inflammatory.  There are many other potential reasons for potential antibiotic residues in food products that aren’t “irrational”.  Strongly recommend “Inappropriate us of antibiotics in food animals is one factor associated with the presence of violative antibiotic residues in food animal products.”  This same sentiment is repeated in lines 44-45, so I would recommend removing there so as not to repeat.

Response: Thank you so much for your valuable comments and suggestion. We have revised the sentence in the revised version of the manuscript (Page 1, lines 29-31).

Line 58 – no need to repeat the (IGS) abbreviation here.

Response: We have removed it in the revised version of the manuscript (page 2lines 60-61).

Line 75-77 – this study does NOT add any information regarding whether violative levels of antibiotics are increasing or decreasing in tissues of food animals sold in grocery stores as this was not the sample tested.  The second half of the sentence after the comma referencing that should be removed.

Response: Thank you for the comment and suggestion. We have revised the statement in the revised version of the manuscript (Page 2, lines 79-83).

M&M:

Lines 89-106 – The methodology used to collapse the data is extremely flawed.  The collapsing of all cattle data into a single variable, and all other species into a separate variable demonstrates a lack of understanding of these production systems and how the use of the VFD, types/routes of antibiotics frequently used, and differences in the levels of violative residues for the antibiotics tested differs between these species.  The cattle data should have been analyzed based on production class, which is given in the dataset (dairy cow, bull, beef cow, bob veal, heifer).  For example, there is no VFD use of antimicrobials in dairy cattle, therefore, there should be no change in antibiotic use related the implementation of the VFD in this class. Antibiotic use in bob veal calves, beef cattle, and swine, however, was likely to experience significant changes after VFD implementation.  In addition, collapsing all of the other animal species into a single variable does not take into account that for some animal species there is a tolerance set, and for others that don’t have labeled antibiotics (ie – goats) there is no tolerance so any residue is a violative residue.  At minimum these should be analyzed separately based on this information.

Response: Thank you so much for your important comments and suggestions. We have revised the analysis according to the animal production class (bob veal, beef cows, dairy cows, bull, heifer, steer, goat, sheep, swine, and turkey) and presented it in the revised version of the manuscript (Page 3, lines 96-115).

General comment – if the hypothesis was as stated “…that the use of injectable antimicrobial drugs may have increased in food animals in the U.S. after the implementation of the VFD, which could increase the detection of violative levels of residues in tissues of food animals in the U.S.”, why were residues for the most common classes of injectable antimicrobials not analyzed (ie – desfuroylceftiofur in dairy cattle; florfenicol, tilmicosin, tulathromycin in beef cattle, etc)?  This suggests a lack of understanding of production practices and antibiotic use in food producing animals in the US, as penicillin, oxytetracycline and sulphonamides are not the most commonly used injectable products.

Response: Thank you so much for your important comments and suggestions. We have included desfuroylceftiofur florfenicol, tilmicosin, and neomycin in our analysis in the revised version of the manuscript (lines 98, 123, 156, 166-173, 230-38, 240-46, 327-333, 334-341). However, we did not find any statistically significant association between predictors and these antibiotics as outcome variables except neomycin was associated with animal production class. We reported these results as text in the main text of the manuscript, and the tables have been presented as supplementary tables (Tables S1 to S8) in the revised version of the manuscript. We did not include tulathromycin as there were very few records in the dataset.

Results:

Tables 1-3 – why was cattle used as referent for first two, and other used as referent for the other?

Response: We have analyzed animal variable as animal production class as you suggested in your earlier comment. We have used -dairy cow- as the reference group in the regression model as there is no VFD use of antimicrobials in dairy cattle.

Discussion:

Line 267-269 – this sentence is not complete, needs edited. “Secondly, while animal and tissue type were significantly associated with penicillin residue violation, VFD implementation did not impact penicillin residue violation.”  Also need to edit 271-273.

Response: We have revised these sentences in the revised version of the manuscript (Line 347-357).

Line 277-279 – The second sentence overreaches in regards to the the results of this study.  As the authors only looked at a small number of antibiotics, primarily older classes of drugs that are not necessarily the most commonly used antimicrobials on farms, they cannot holistically state that the VFD did or did not result in an increased used of injectable antibiotics.  Recommend removal of second sentence. 

Response: Thank you for the suggestion. We have revised these statements in the revised manuscript (lines 359-68), including removing the sentence as suggested.

Paragraph 287-208 – All veterinary-directed penicillin use in cattle is extralabel as the labeled dose is not effective.  This is not currently discussed but is an important part of the discussion that should be included, particularly as the original label on the bottle which is available over the counter currently lists a much shorter withdrawal. This discrepancy likely leads to errors in withdrawal time.  All of the antibiotics selected for study in this paper are currently available as OTC antibiotics, which may play a key role in the risk of residues, yet this is not discussed anywhere in the paper – this needs to be added.   Clinical illness can also impact the withdrawal time (as these are established in healthy animals), and may also play a role in these violative residues, and was not discussed

Response: Thank you so much for the comments and suggestions. We have revised these issues in the revised version of the manuscript (lines 370-86).

Paragraph 325-335 – This entire paragraph is very problematic.  Was the level required for violative residue considered in the model?  If no label, then any residue = violative.  Minor animal species have very few labeled drugs (ie – goats only have ceftiofur, sheep only ceftiofur, tilmicosin, and oxytetracycline).  See comment from M&M above - I do not think it was appropriate to lump all of these non-cattle species together for the analysis because of this issue.  Statement 328-330 is incorrect as written and should be removed or reworded.  Extralabel use in these species (ie - sheep and goats) is common simply because there is no labeled products, so it is a necessity that is perfectly legal.  As currently written, the results of this study should be a call for increased labeled antibiotic options for these animals which would then provide producers with appropriate withdrawal times to follow and establish tolerance levels.  333-334 is also an incorrect statement as extralabel drug use is perfectly legal if done under a valid VCPR, and the veterinarian of record is responsible for establishing a withdrawal.  Please reword to reflect this, or remove.

Response: Thank you so much for the comments and suggestions. We have revised this paragraph to address these issues in the revised version of the manuscript (lines 416-436).

Overall recommendations:

While the study seeks to address an important question regarding antimicrobial use and residues following implementation of the VFD, the current data analysis has significant flaws as stated above, and is not publishable in its current state because of these flaws.  To address these issues, the data analysis could be re-run taking into consideration the issues identified above (preferred method).  The authors could instead attempt to clearly articulate the flaws in their study design in the discussion section, however, this is not the preferred method and may not be sufficient to address concerns identified.  It is also recommended that the authors seek out additional expertise in pharmacology and production animal medicine to assist them in their analyses given the concerns that were identified.

Response: Thank you so much for your thorough review, comments, and suggestions. We believe your comments and suggestions substantially improved the revised version of the manuscript.

Round 2

Reviewer 2 Report

It's OK.

Author Response

Thank you so much, for your comment. 

Reviewer 6 Report

Thank you for the corrected analysis to address the animal class and the addition of other important antimicrobials.  I believe this has improved the paper substantially.  However, after reading the new analysis, there remains a very significant issue with how the odds ratio results are being reported.  Throughout the paper the ORs are reported as “association between predictors and detection of violative residues”, however, I believe you would need to know the total number of “exposed” animals (ie – number of animals slaughtered before and after VFD) to be able to say if there were actually more or less actual “cases” of violative residues and decreased odds of that occurrence in the total population.  What it appears is being reported instead is the odds of a violative residue compared only to a non-violative residue from the IGS samples.  As non-violative residues are not a food safety concern and are likely driven by other factors (number of suspect animals pulled, etc), this does not seem to be an appropriate comparison and does not truly indicate if the odds of a violative residue increased or decreased in the entire population – it is only indicative of the IGS samples.

Therefore, in every location in the paper that says “association between predictors and detection of violative residues” it should be clarified as “association between predictors and detection of violative residues compared to non-violative residues in IGS samples.”  There are many locations where this will need to be corrected, and the authors will also need to review their discussion and findings with this in mind and ensure that the conclusions are restricted to this dataset, and not assumed to refer to the entire population as a whole.   This also heavily affects the production class analysis as previously pointed out, violative residues are more likely in animal classes where the antibiotic in question has no label, therefore all residues detected are typically considered a violation as there is no tolerance level set.

Additional recommendations:

Title:  Should be clarified to the following “Effect of changes in veterinary feed directive regulations on violative versus non-violative antibiotic residues in the tissue of food animals in the United States” based on the data that was utilized in this study (see discussion above).

Hypothesis:  Currently states “…we hypothesize that the use of injectable antimicrobial drugs may have increased in food animals in the U.S.; neomycin is not an injectable antibiotic and is only used as an oral formulation.  Thus, I would recommend removing from the paper if the authors wish to focus on injectable medications.

For all analyses, dairy cows are used as the reference population.  Cull dairy cows are one of the most likely production classes to have residues, but this is not consistent for all drug classes.  The authors should ensure that this is clearly stated in the manuscript, and clarify where appropriate if this impacts the analysis.

Line 172 – florfenicol spelled incorrectly

Line 367-368 – This is an incorrect statement in regard to penicillin and withdrawal time. While the label for penicillin G has a short withdrawal time (either 4 or 10 days), the labeled dose is not effective and therefore all penicillin use under the direction of a veterinarian is at extra-label dosing which requires an extended withdrawal time (see FARAD digest for more information: http://www.farad.org/publications/digests/112006ExtralabelPenicillin.pdf , typically at least 21-30 days depending on dose/route/duration).  This should be discussed/clarified.

Paragraph 371-397 – This paragraph also still discusses penicillin and contradicts the decrease in penicillin violative residues that was identified.  The authors should re-evaluate this paragraph after making the changes recommended above.

Lines 384-387 – The addition of this statement would have been correct if the authors only kept the original antibiotics.  Penicillin, oxytet, and sulfas are available over the counter.  Ceftiofur, florfenicol and tilmicosin are not.

Author Response

Comments and Suggestions for Authors

Thank you for the corrected analysis to address the animal class and the addition of other important antimicrobials.  I believe this has improved the paper substantially.  However, after reading the new analysis, there remains a very significant issue with how the odds ratio results are being reported.  Throughout the paper the ORs are reported as “association between predictors and detection of violative residues”, however, I believe you would need to know the total number of “exposed” animals (ie – number of animals slaughtered before and after VFD) to be able to say if there were actually more or less actual “cases” of violative residues and decreased odds of that occurrence in the total population.  What it appears is being reported instead is the odds of a violative residue compared only to a non-violative residue from the IGS samples.  As non-violative residues are not a food safety concern and are likely driven by other factors (number of suspect animals pulled, etc), this does not seem to be an appropriate comparison and does not truly indicate if the odds of a violative residue increased or decreased in the entire population – it is only indicative of the IGS samples.

Therefore, in every location in the paper that says “association between predictors and detection of violative residues” it should be clarified as “association between predictors and detection of violative residues compared to non-violative residues in IGS samples.”  There are many locations where this will need to be corrected, and the authors will also need to review their discussion and findings with this in mind and ensure that the conclusions are restricted to this dataset, and not assumed to refer to the entire population as a whole.   This also heavily affects the production class analysis as previously pointed out, violative residues are more likely in animal classes where the antibiotic in question has no label, therefore all residues detected are typically considered a violation as there is no tolerance level set.

Response: Thank you so much for your important comment and suggestions. We totally agree with your comments and suggestions. Our study samples were belonging to the inspector-generated sampling (IGS) samples, not the general animal population in the slaughterhouse. Also, we have amended the statement related the outcome variable “The outcome of interest for each antibiotic or antibiotic group was whether violative residue was present compared to absence in the tissue of food animals from the IGS.” in the methods and materials section of the revised manuscript (lines: 118-19). Therefore, we did not repeat the word compared to non-violative residues in other places of the manuscript while reporting the odds ratio (OR).

Besides, we have revised the statements related to IGS samples in the discussion section of the revised version of the manuscript “The result of this study would not be generalizable because the tissue samples were collected using the targeted sampling of food-producing animals under the IGS. The tissue samples from the IGS were chosen based on clinical signs or pathologic lesions on food-producing animals during antemortem and post-mortem examination by an authorized veterinarian for collecting the tissue samples. So current study results may over-represent the violation of antibiotic residues in tissues of food-producing animals from the IGS than all other food-producing animals brought to the slaughter-house. Hence, the results of the current study should be generalized cautiously because of the potential sampling bias. Our study findings only apply to the samples collected under the IGS, not the entire food-producing animals brought into the slaughterhouse.” (lines:454-463). Additionally, we have already described the methods (briefly) of inspector-generated sampling under the National Residue Program in the introduction section of the manuscript (line: 55-69).  

Additional recommendations:

Title:  Should be clarified to the following “Effect of changes in veterinary feed directive regulations on violative versus non-violative antibiotic residues in the tissue of food animals from IGS in the United States” 

Response: Thank you so much, for your suggestion. We have revised the title as “Effect of changes in veterinary feed directive regulations on violative antibiotic residues in the tissue of food animals from the inspector-generated sampling in the United States”

Hypothesis:  Currently states “…we hypothesize that the use of injectable or parentally antimicrobial drugs may have increased in food animals in the U.S.; neomycin is not an injectable antibiotic and is only used as an oral formulation.  Thus, I would recommend removing from the paper if the authors wish to focus on injectable medications.

Response: Thank you so much for pointing out this issue. We have removed the neomycin related information from the revised version of manuscript as we focus on the injectable antibiotics.

For all analyses, dairy cows are used as the reference population.  Cull dairy cows are one of the most likely production classes to have residues, but this is not consistent for all drug classes.  The authors should ensure that this is clearly stated in the manuscript, and clarify where appropriate if this impacts the analysis.

Response: We have discussed this issue in the discussion section of the revised version of the manuscript (lines:469-475).

Line 172 – florfenicol spelled incorrectly

Response: We have corrected the spelling in the revised version of the manuscript (line 176).

Line 367-368 – This is an incorrect statement in regard to penicillin and withdrawal time. While the label for penicillin G has a short withdrawal time (either 4 or 10 days), the labeled dose is not effective and therefore all penicillin use under the direction of a veterinarian is at extra-label dosing which requires an extended withdrawal time (see FARAD digest for more information: http://www.farad.org/publications/digests/112006ExtralabelPenicillin.pdf , typically at least 21-30 days depending on dose/route/duration).  This should be discussed/clarified.

Response: Thank you so much for pointing out this issue. We have corrected this issue and discussed in the revised version of the manuscript (lines:) Besides, we have cited the reference (http://www.farad.org/publications/digests/112006ExtralabelPenicillin.pdf) in the revised version of the manuscript [lines:373-383].

Paragraph 371-397 – This paragraph also still discusses penicillin and contradicts the decrease in penicillin violative residues that was identified.  The authors should re-evaluate this paragraph after making the changes recommended above.

Response: Thank you for your comment. We have corrected this paragraph in the revised version of the manuscript [lines: 384-405].

Lines 384-387 – The addition of this statement would have been correct if the authors only kept the original antibiotics.  Penicillin, oxytet, and sulfas are available over the counter.  Ceftiofur, florfenicol and tilmicosin are not.

Response: Thank you so much for your valuable comment and suggestion. We have revised this in the revised version of the manuscript [lines:397-405]. We are grateful to you again for your valuable comments and suggestions. We strongly believe your comments and suggestions improved considerably in the revised version of the manuscript.